# Gender Differences and miRNAs Expression in Cancer: Implications on Prognosis and Susceptibility

**DOI:** 10.3390/ijms241411544

**Published:** 2023-07-17

**Authors:** Santino Caserta, Sebastiano Gangemi, Giuseppe Murdaca, Alessandro Allegra

**Affiliations:** 1Division of Hematology, Department of Human Pathology in Adulthood and Childhood “Gaetano Barresi”, University of Messina, Via Consolare Valeria, 98125 Messina, Italy; 132588@polime.it (S.C.); aallegra@unime.it (A.A.); 2Allergy and Clinical Immunology Unit, Department of Clinical and Experimental Medicine, University of Messina, Via Consolare Valeria, 98125 Messina, Italy; gangemis@unime.it; 3Department of Internal Medicine, University of Genova, Viale Benedetto XV, 16132 Genova, Italy; 4IRCCS Ospedale Policlinico San Martino, 16132 Genova, Italy

**Keywords:** sex differences, male, female, X chromosome inactivation, genomic imprinting, epigenetics, miRNA, cancer, apoptosis, oncoMir

## Abstract

MicroRNAs are small, noncoding molecules of about twenty-two nucleotides with crucial roles in both healthy and pathological cells. Their expression depends not only on genetic factors, but also on epigenetic mechanisms like genomic imprinting and inactivation of X chromosome in females that influence in a sex-dependent manner onset, progression, and response to therapy of different diseases like cancer. There is evidence of a correlation between miRNAs, sex, and cancer both in solid tumors and in hematological malignancies; as an example, in lymphomas, with a prevalence rate higher in men than women, miR-142 is “silenced” because of its hypermethylation by DNA methyltransferase-1 and it is blocked in its normal activity of regulating the migration of the cell. This condition corresponds in clinical practice with a more aggressive tumor. In addition, cancer treatment can have advantages from the evaluation of miRNAs expression; in fact, therapy with estrogens in hepatocellular carcinoma determines an upregulation of the oncosuppressors miR-26a, miR-92, and miR-122 and, consequently, apoptosis. The aim of this review is to present an exhaustive collection of scientific data about the possible role of sex differences on the expression of miRNAs and the mechanisms through which miRNAs influence cancerogenesis, autophagy, and apoptosis of cells from diverse types of tumors.

## 1. Introduction

### 1.1. General Considerations on miRNAs

MicroRNAs are small, noncoding molecules of about twenty-two nucleotides and have crucial roles in both healthy and pathological cells; in fact, they are involved in various intra- and intercellular mechanisms like proliferation and apoptosis, so their incorrect function is involved in the onset of different diseases, such as cancer [1,2]. In detail, a miRNA could have a lot of genes as targets and, at the same time, a gene could be regulated by different miRNAs [3]; according to the scientific literature, they control about 60% of human genes [4]. miRNAs derive from a long molecule of ribonucleic acid called pri-miRNA that is caped and polyadenylated in the nucleus from polymerase II and then processed from DiGeorge syndrome critical region 8 (DGCR8) and Drosha proteins to form the pre-miRNA, consisting of about seventy nucleotides. Next, this molecule moves to the cytoplasm through the exportin 5–RAs-related nuclear protein (RAN) complex and is cut from an endonuclease into the mature miRNA. They can be secreted into vesicles called exosomes in the extracellular microenvironment or circulate tied to lipoproteins and argonaut proteins [3] and act by tying to the 3′ untranslated regions of mRNA and, consequently, blocking its translation or promoting protein degradation [5,6]. Some miRNAs are expressed in certain tissues and in a specific sex; for example, molecules from the miR-35 family are involved in sex determination, while miR-532 and miR-660 are highly expressed but lowly methylated in females rather than males [3,6,7]. Furthermore, in the X chromosome, there are a lot of miRNAs, while in the Y chromosome, there are only a few [8]. In addition to genetic mechanisms, epigenetics regulate their expression also [3]. In fact, external factors like infections, food, traumatic experiences, chemical pollutants, cold, and heat lead to changes in miRNA expression that can be transmitted to the next generations. Epigenetic mechanisms include the X chromosome inactivation in females and genomic imprinting. It is well known that, during embryo development, the inactivation of one of the two X chromosomes occurs in order to maintain a balanced expression of X chromosomes with men having just one; an incomplete inactivation of the X chromosome leads to a biallelic expression of miRNA [8]. The genomic imprinting consists of the expression of just one allele in all the genes of a subset of maternal and paternal origin, localized in the so-called “Differentially Methylated Regions” of the human genome. A particular type of imprinting is “hormonal imprinting”, a mechanism during which the hormonal receptor ties for the first time with a hormone, and the latter becomes its life-long target; however, during endocrine system genesis, receptors do not have an absolute specificity, so synthetic molecules like drugs or other hormones similar to the target one could tie to the receptor, leading to wrong hormonal imprinting and, consequently, to life-long events transmitted to future generations [7]. Cancer cells present alterations in the expression of genes related to molecules that regulate proliferation, death, and survival, including miRNAs. In this context, tumor-suppressor micro-RNAs (oncoMirs) have the capacity to promote cancerogenesis by suppressing tumor-suppressor genes and, consequently, inhibiting cell-death pathways like apoptosis; they are upregulated in tumor cells and in the tumor microenvironment. For example, in B-cell lymphoma, miR-17-92 inhibits the “pro-apoptotic protein Bcl-2-like protein 1” (Bcl-2L1), inducing resistance to apoptosis and promoting the proliferation of immature lymphoid cells. On the other hand, antagoMirs are synthetic miRNAs, antagonists first developed as silencing molecules of miRNAs in the first years of this millennium; in detail, they are antisense oligonucleotides (ASOs) that tie to specific ribonucleic acid (RNA) sequences and, by inducing RNase H-mediated cleavage, influence the expression of specific miRNAs, leading to a reduction of cancer-related ones [9].

### 1.2. Sex Differences, miRNAs, and Cancer

MiRNAs have several roles in cancer also related to sex chromosomes. In fact, the X chromosome contains a high density of these small noncoding molecules involved in the regulation of immune response in humans, specifically, of immunosurveillance against the onset and progression of cancer [8]. The immune system has demonstrated roles also in cancer therapy as shown by a new class of drugs called “immune checkpoint inhibitors” [10]. We know tumor cells have the ability to escape from immune recognition as nonself-factors through the action of molecules like programmed cell death protein 1, a receptor that is tied by its proteic targets programmed death-ligand 1 (PD-L1) and programmed death-ligand 2 (PD-L2) with the result of an inhibition of T cells and the consequent block of apoptosis of cancer cells. However, PD-L1 expression is regulated by X-linked miRNAs such as miR-106b and miR-20b. The scientific literature showed their oncogenic function and their ability, especially for miR106a, in downregulating the anti-inflammatory activity of cytokine IL-10. Other miRNAs involved in potentiating inflammation with an obvious stimulus to cancerogenesis are miR-18 and -19, since they act on the nuclear factor-κB (NF-κB) [8,11,12]. Furthermore, miRNAs can also be secreted by the pituitary gland and, since the last is dependent on sex hormones, menopause in women and other hormonal changes in men conduct sex-related changes in miRNA production from the pituitary gland. In particular, an increase in proapoptotic miRNAs and a decrease in antiapoptotic ones verifies in this sex-related manner leading to an inhibition of all stimulating mechanisms, such as proliferation and neoangiogenesis, and to the induction of apoptosis, with a possible antitumoral effect [6]. In some cases, scientists succeeded in identifying the role of specific miRNAs; for example, the capacity of miR-22 to act on methylenetetrahydrofolate dehydrogenase 2 and methylenetetrahydrofolate reductase blocking and, consequently, the S-adenosylmethionine synthesis with the result of cancer cell death. Moreover, in females but not in males, an important expression of miR-27a-3p has been observed in cases of high tumor stage, while the expression of miR-17-5p and miR-20a-5p has been observed to be reduced in metastatic cancer cells [13]. The onset of cancer was recently associated with the so-called “XCI genes”, genes that escape from X chromosome inactivation with the consequent failed balance between male and female genomic balance; among them, researchers found that the connector enhancer of kinase suppressor of ras 2 (CNKSR2), lysine demethylase 6A (KDM6A), alpha-thalassemia/mental retardation, X-linked (ATRX) and Lysine-specific demethylase 5C (KDM5C) are mutated in males with cancer but not in females. On the other hand, toll-like receptor 7 (TLR7), chromosome X open reading frame 21 (CXORF21), and CD40L are double expressed in women, determining an augmented risk of autoimmune diseases [9]. The scientific literature demonstrates that miRNA expression can be influenced by natural substances such as genistein, a molecule belonging to the chemical class of isoflavones, with described activities of phytoestrogen and of an angiogenesis inhibitor; it can be found in different vegetables like fava beans, soybeans, lupin, and coffee. Experiments conducted using genistein put in evidence its capability to cause in vitro the stop of the cell cycle at the G2/M phase by interacting with proteins involved in cell proliferation and cancer growth such as kinesin family member 20 (KIF20) of the kinesins family. Western blot analysis showed which genistein is capable of inhibiting the translation of the (Cdc25C) protein, cyclin-dependent kinase 1 (CDk1), cyclin A, and cyclin B, and stimulating the expression of cyclin-dependent kinase (CDK) inhibitor and cyclin-dependent kinase inhibitor 1 (p21 CIP1/WAF1); furthermore, it modulates the RAS/RAF pathway, stabilizing activation and phosphorylation of MAPK and inhibits the Akt and JAK/STAT pathways, both crucial for cell survival. miRNA expression can be influenced by genistein; in fact, it has been shown that this natural substance is capable of downregulating miR-151, responsible for invasion and cell migration in prostate cancer, and the minichromosome maintenance (MCM) gene family, involved in DNA replication and cancerogenesis. However, there are obstacles to the daily use of genisteins because of their poor water solubility and low serum availability; thus, solid–lipid–particulate systems (SLPs) have been proposed. In fact, these carriers consent to the drug being dissolved, encapsulated, and attached to a nanoparticle matrix. For example, a combination of genistein and doxorubicin has been associated with lower adverse effects compared with doxorubicin hydrochloride alone and induces a decrease in ROS production by prostate cancer cells [14,15].

In this review, we presented a collection of scientific data about the possible role of sex differences on the expression of miRNAs and the mechanisms through which miRNAs influence the cancerogenesis, autophagy, and apoptosis of cells from diverse types of tumors. We researched data on PubMed using the keywords “Sex differences”, “Male”, “Female”, “X chromosome inactivation”, “Genomic imprinting”, “Epigenetics”, “miRNA”, “Cancer”, “Apoptosis”, “oncoMir” in a period of time from 2006 to 2023.

## 2. Hematological Malignancies

Hematopoietic stem cells derive from the mesoderm during embryogenesis and are crucial for the correct development of blood cells and the functioning of the immune system [15,16,17]; both genetic and environmental factors such as chemical substances, radiation, and infections can cause their dysfunction and lead to hematological malignancies. Hematological malignancies, like leukemia, lymphoma, and myeloma, represent 7% of new cancer cases globally every year and, despite new treatments including drugs and bone marrow transplantation, they have survival rates far from 100% [18].

Studies demonstrated that miRNAs are relevant in the determination of hematological diseases and their complications [19,20].

Hematological malignancies show larger rates of incidence and mortality in men than in women. For example, there is a 6.1% risk of acute myeloid leukemia and a 4.3% mortality rate in males vs. the 4.2% risk and mortality rate of 2.8% in females; the same sex correlation can be observed for myelodysplastic syndromes and for lymphomas (especially for mantle-cell type and less for marginal zone lymphoma). Therefore, it can be stated that sex could be considered a negative prognostic factor both in the onset and progression of hematological malignancies [21]. Studies of molecular biology demonstrated that miR-342, localized in the Enah/Vasp-Like (EVL) gene of chromosome 14, regulates cytoskeleton remodeling and, consequently, the capacity of migration of cells. In B-cell lymphomas, miR-342 is silenced because of the hypermethylation of the EVL gene in its promoter region and the treatment with decitabine leads to the re-expression of miR-342 and the translation of the EVL protein. It is well known which DNA methyltransferase-1 normally determines the hypermethylation of genes; in this case, miR-342 leads to a downregulation of this enzyme and, consequently, to the hypomethylation of cells and a reduction in B-cell lymphoma aggressiveness [22]. Furthermore, men have a worse outcome in terms of systemic effects like sarcopenia than women and this condition can be related to the fact that hormones such as estradiol in females reduce the viscosity of the mitochondrial membrane, improving the activity of the skeletal muscle; in this mechanism, miR-486 is involved and supports the synthesis of myotubes from myoblasts, so it can be considered a sex-related biomarker of cancer systemic effects [23] (Figure 1).

### 2.1. Thyroid Cancer

Papillary thyroid carcinoma (PTC) is the most frequent endocrine tumor with an incidence rate from two to four times higher in women than men and a good prognosis except for the 10% of patients who develop metastases; conventionally, the diagnosis is made with ultrasound, which consents to the detection of thyroid nodules; another method is the fine needle aspiration biopsy with the guide of ultrasound [24,25,26,27,28,29]. It has been related to genetic mutations of the rearranged during transfection (RET) gene and papillary thyroid cancer/rat sarcoma viral oncogene homolog/*B-Raf* murine sarcoma viral oncogene homolog B1 (PTC-RAS-BRAF) pathway, with the last altered in most patients. In cells from papillary thyroid cancer, an overexpression of miR-221, miR-222, miR-181a and miR-181b, and miR-21 has been shown compared to the expression in normal thyroid cells and the mechanism proposed is that of regulation of the cell cycle throughout a modulation in the expression of the protein p27Kip1; at the same time, it is possible to put in evidence a downregulation of miR-345, miR-219, and miR-26a; in order to evaluate the role of miR-219-5p in papillary thyroid cancer, researchers studied the expression of different miRNAs in thirty PTC samples and thirty tissues from normal thyroids using quantitative polymerase chain reaction and demonstrated that there were no differences between the two groups in age and sex; furthermore, tumor size was shown to be larger and lymph node metastases were more frequent in cases with low expression of miR219-5p [28]. Papillary thyroid cancer is also influenced by the hormonal situation of the patient. In fact, it has been demonstrated that 53% of cases of papillary thyroid carcinoma are estrogen-receptor (ER) positive; the estrogen-receptor pathway, which also includes extracellular signal-regulated kinases (ERK) and *p38* mitogen-activated protein kinases, is a class of mitogen-activated protein kinases (p38 MAPK) that induce proliferation and migration of cancer cells via the action of miR-219-5p and promote metastasis spread [30] (Figure 2).

### 2.2. Hepatocellular Carcinoma

Hepatocellular carcinoma (HCC) is among the most malignant tumors and comprises primary and secondary forms. In addition to advantages in diagnostic strategies and new treatments, the prognosis of HCC is not satisfying. The first trigger in Asia is chronic hepatitis B virus infection, while in Western countries the main causes are chronic hepatitis C virus, alcohol consumption and cirrhosis, steatohepatitis, and diabetes; it has a 3–5 higher incidence in males than in females. In addition to environmental factors, the expressions of specific miRNAs and sex hormones influence the onset and progression of this tumor [31,32,33,34,35,36,37,38,39,40]. The family of miR-371-373 includes miR-371a-5p and is located in chromosome 19; these miRNAs are overexpressed in stem cells during embryogenesis and then quickly reduce after cell differentiation. In HCC, the “silencing” of miR-371a-5p does not verify and it is present at high levels in these adult patients, especially in men. It acts to promote the transition from the G to S phase of the cell cycle, the tumor growth in volume and weight, neovascularization in vivo, and inducing the proliferation of cells, migration, colony formation, and invasion in vitro. MiR-371a-5p is a component of the pathway, including lymphoid enhancer-binding factor 1/SRC kinase signaling inhibitor 1/pleiotrophin/snail family transcriptional repressor 2 (LEF-1/SRCIN1/PTN/Slug), cause of which testosterone induces an upregulation of miR-371a-5p leading to proliferation, neoangiogenesis and metastasis of HCC, especially in men. A poor prognosis and lower overall survival rates in patients with significant levels of miR-371a-5p have been observed; so, it could be considered a good biomarker in HCC [41]. In women with HCC, instead, it has been identified as an overexpression of miR-18a, with a crucial role for p53 in the process leading from pri- to pre-miR-18a. The pathway p53/miR-18a/ER acts just in cells of hepatocellular carcinoma and not in precancerous conditions; moreover, it occurs in young women. Thus, this pathway contributes to carcinogenesis before menopause [42]. In addition, miRNAs participate in the response to treatment in HCC; in fact, therapy with estrogens determines an upregulation of the oncosuppressors miR-26a, miR-92, and miR-122 and a downregulation of the oncomiRs miR-143 and miR-17 [43].

### 2.3. Colorectal Cancer

Colorectal cancer (CRC) is among the most common cancer throughout the world; it has a higher prevalence in men than women, with some exceptions. In fact, cancer affecting the ascendent tract of the colon is more frequent in women; moreover, sex differences are part of the prognosis of this tumor since young women up to forty-four years old have a better prognosis than young men and women in their postmenopausal period. Long-term colorectal inflammation, for example, due to Crohn’s disease and ulcerative colitis, together with a dysregulation of the immune response, leads to an augmented risk of this type of cancer. Since it is well-known that autophagy has a crucial role in cell mechanisms of life and death, in colorectal tumors, it has been possible to identify some miRNAs with an inhibiting effect on autophagy via a wrong regulation of “autophagy-related proteins” (ATG) and, consequently, a negative influence on the immune response to nonself-cells like neoplastic ones. For example, miR-142-3p targets the mRNA of IL-8 and inhibits the ATG “inflammatory bowel protein 1” [4]. MiRNAs expression is influenced by sex hormones; in fact, ERβ ligation causes the arrest of the cell cycle at the G1-S phase and induces autophagy via BNIP3, a molecule of the Bcl-2 family that modulates the permeability of mitochondrial membranes. MiRNAs are also involved in the response to cancer treatment, in particular, miR-22 acts on estrogen receptor-β (Erβ) mRNA causing the inhibition of estrogens activity and, consequently, inducing an augment of cancer cells’ sensitivity to 5-fluorouracil, a drug commonly used in CRC; moreover, miR-22 regulates the so-called “B-cell translocation gene 1” that, in turn, modulates cell differentiation and growth [22]. The expression of miRNAs is different in males and females; in fact, higher levels of miR-16, related to a worse survival rate but, on the other hand, downregulated in CRC tissues, are present in men than women and are associated with advanced stages of the disease [4].

### 2.4. Gastric Cancer

Gastric cancer (GC) is a multifactorial tumor formed from epithelial cells of gastric mucosal and represents the fourth tumor for incidence and the fifth cause of cancer-related death all over the world, with a five-year survival rate that is not more than 40%. A small quantity of reactive oxygen species (ROS) is normally produced in healthy organisms and they have important roles in cell signaling and immune responses to pathogens but cause excessive production of ROS; so, prolonged oxidative stress causes damage to cell membranes, macromolecules, and DNA, activating inflammation pathways and leading to different diseases [44]. In addition to the improvement of endoscopic techniques, the diagnosis is often made at late stages; however, early gastric cancer is potentially curable with similar survival rates of endoscopic resection compared to the surgical one. Actually, the treatment consists of chemotherapy, radiotherapy, surgery, or combined targeted therapy. The onset of gastric cancer is influenced by both genetic and environmental factors such as food and alcohol consumption with a double incidence in men than women [45,46,47]. MiRNAs could have a role in cancerogenesis because miR-125b directly targets proapoptosis genes blocking their function and leading to uncontrolled cell proliferation; as far as gastric cancer treatment is concerned, in vivo and in vitro experiments have been conducted using bicalutamide, an antagonist of androgen receptor involved in the pathogenesis of the disease. Researchers demonstrated that, while the treatment of cells with androgen for twenty-four hours importantly led to an augment of miR-125 expression, after an administration of 40 μM of bicalutamide singly or combined with 50 nM of dihydrotestosterone, there was an important decrease in the expression of miR-125 in cases than in controls, put in evidence by a green fluorescent signal in the nucleus and diffused cytoplasmic staining at the microscopic observation. Therefore, it can be stated that bicalutamide is able to downregulate the androgen receptor (AR)/miR-125 pathway and induce apoptosis of cells from gastric cancer [48].

### 2.5. Lung Cancer

Lung cancer is commonly divided into nonsmall-cell lung cancer (NSCLC) and small-cell lung cancer (SCLC). NSCLC represents 80% of lung tumors; it is often diagnosed at advanced stages with the consequences of poor prognosis and high mortality rates all over the world. From the discovery of immunotherapy, with a single drug or in combination, a new era against cancer started. As explained for hematological malignancies, also in solid tumors immune checkpoint inhibitors, directed, for example, against programmed death protein-1 or the cytotoxic T lymphocyte antigen 4, are suitable permitting the reaching, in some cases, of superior results in terms of overall survival [49,50,51,52,53,54,55,56,57,58,59]. Oxidative stress, as is well known, has a pivotal role in cancerogenesis, in particular, the nuclear factor erythroid 2-related factor 2/*heme oxygenase*-*1* (Nrf2/HO-1) pathway is involved in chemoresistance in lung cancer through a stimulus to proliferation; in fact, an overexpression of HO-1 is related to higher tumor aggressivity [60]. In the last thirty years, the incidence of lung carcinoma importantly decreased in men while augmented in women, especially in those in their premenopausal period. In fact, they are frequently affected from an advanced stage, with less differentiated types of cancer and having a more elevated number of metastases compared with men or postmenopausal women. This condition can be explained considering the hormonal effects of estrogen, but also with an incorrect expression of specific miRNAs. It is well-known that estradiol stimulates the expression of cyclin D and c-myc that, in turn, lead to the progression of the cell cycle; moreover, the oncogene FAT is, importantly, reduced in nonsmall-cell lung cancer than adjacent normal tissues and it is demonstrated to be capable of causing autophagy of tumor cells. Among miRNAs involved in the pathogenesis and response to therapy of lung cancer, there is miR-153-3p, a molecule lowly expressed in these cases of nonsmall-cell lung cancer resistant to the epidermal growth factor receptor (EGFR) tyrosine kinase inhibitor. Interestingly, miR-153-3p levels are correlated with an important presence of ATG5 protein which leads to apoptosis due to its inhibitory effect on protective autophagy of cells from lung carcinoma [22]. In vitro and in vivo studies demonstrated the suppressive effect on cell growth of miR-143 and miR-145; in particular, when expressed in stromal cells, they play a protective role, while a low stromal expression is related to poor survival rates [61].

### 2.6. Melanoma

Melanoma is the most aggressive skin cancer, with an incidence a little higher for males than females (1:66 vs. 1:85). Lifestyle has a crucial role in cancerogenesis since ultraviolet exposure is among the most important risk factors. Also, there is a typical distribution of melanomas between the two sexes with a prevalence of extremities involvement in women and of the back in men, the condition of the latter which makes difficult a self-detection of the lesions. Ultraviolet radiation exposure causes an augment in ROS production; at the same time, the activation of Nrf2 leads to an antioxidant cellular response that protects against ultraviolet radiation toxicity. If Nrf2 develops a gain-of-function mutation, the risk of melanoma increases, suggesting a possible role in cancerogenesis. Keratinocytes and melanocytes, after ROS stimulation, produce proinflammatory cytokines which, together with nonimmune molecules such as growth factors, influence innate and adaptive immune response and promote tumorigenesis. In this context, mast cells are recruited, followed by monocytes/macrophages that migrate to the site of proliferation, producing cytotoxic molecules against cancer cells. As it is of note, the woman’s immune system is more efficient compared with the male one and is capable to organize a stronger innate and adaptive response. These differences are controlled by a complex network of signals, including that coming from miRNAs. It has been demonstrated that miR-221 and miR-222 have a suppressive effect on the proliferation and dissemination of cells from melanoma since they target specific molecules like the protoncogenes c-kit and ETS proto-oncogene 1, transcription factor (ETS-1) [7]. Since autophagy is important to prevent the negative effects of ultraviolet irradiation for the epidermis, an altered expression of miR-23a causes premature senescence of fibroblasts in the skin and, consequently, the block of normal, positive autophagy; however, the antagomiR-inactivation of miR-23a causes a cascade of stimulus, leading, in turn, to ultraviolet-dependent autophagy with all positive consequences in preventing fibroblasts from precocious senescence [8,22].

### 2.7. Breast Cancer

Breast cancer is the second cause of cancer death in women all over the world and it has recognized risk factors of diet, lifestyle, and genetic mutations. On the one hand, the reduction in alcohol consumption and smoking led to a lower breast cancer risk but it has to be considered that food could have a crucial role; In fact, studies put evidence that the Mediterranean diet, including milk products, soy products, and fibers, could have a protective role against breast cancer onset. Obviously, all of this data needs to be clarified because it has to be specified that food from the Mediterranean Sea has been linked to an augmented risk of cancer due to some chemical components. For example, it could contain different pollutants and compounds, including estrogens which can cause an increased risk of DNA abnormalities and mitotic activity, leading, in turn, to an uncontrolled proliferation typical of cancer [62,63,64,65,66,67,68,69,70,71,72,73,74,75,76,77,78]. Specific molecules such as breast cancer susceptibility gene 1 (BRCA1) and miRNAs permit the characterization of the genetic profile of the disease and a correct screening of patients [79]. Breast cancer constitutes the most common neoplasm in women in Western countries, even if it presents with a low incidence in men; in the last case, but also females, the pathogenesis is frequently associated with a mutation in the oncosuppressor gene BRCA1 and miR-17 and miR-21 have a crucial role in determining this genetic alteration. In detail, they are able to bind to the 3′ *untranslated region* (UTR) sequence of mRNA of BRCA1, causing the inactivation of this gene and, consequently, an uncontrolled cell proliferation [80]. Let-7a miRNA is capable of downregulating estrogen receptors in both men and women and stimulating neoangiogenesis. These miRNAs are overexpressed in all cases of familial breast cancer compared with sporadic ones, while miR-124 results in downexpressed miRNAs, even if its function is not clear yet [81].

### 2.8. Glioblastoma

Glioblastoma is the most aggressive type of glioma with the major prevalence between brain tumors, a poor prognosis, and few therapy perspectives [82,83,84,85,86,87,88,89,90,91,92,93]; for its diagnosis, multiple brain biopsies would be necessary but, since they are invasive procedures, it is impossible. For this reason, it would be very interesting to identify a biomarker that lets clinicians recognize this cancer at a precocious stage of development. Although there are strong standards of cure like chemotherapy and surgical resection, the survival rate is low; for this reason, immunotherapeutic strategies, including immune checkpoint inhibitors, are becoming increasingly attractive for this tumor [10,84]. miRNAs have been related to glioblastoma onset; specifically, hsa-miR-1909-5p, hsa-let-7c-5p, and hsa-miR-206-5p have been shown to have crucial roles in different tumoral mechanisms. Studies demonstrated that hsa-let-7c-5p is involved in the etiology of glioblastoma because it regulates cyclin D1 expression through the Wnt/β-catenin pathway in osteoblasts. Thus, it is a suppressor miRNA with functions in the cancer cell cycle. This is true especially for primary glioblastoma, since in the recurrent forms it is downregulated and, consequently, stimulates Kirsten rat sarcoma virus (K-RAS), PBX homeobox 3 (PBX3), and matrix metallopeptidase 11 (MMP11) with the result of a promotion in cell migration and invasion. Hsa-miR-206 has a suppressive function also. It is able to inhibit cancer-cell migration and induce apoptosis. Unfortunately, in glioblastoma, its expression is often “silenced” with low levels of these small noncoding molecules compared with healthy tissues and, as a consequence, a stimulus for onset and progression. Even if more in vitro and in vivo studies are necessary, a possible biomarker could be hsa-miR-1909-5p since, in most cases of glioblastoma, it is overexpressed [85] (Table 1).

### 2.9. Conclusions

MicroRNAs are small, noncoding molecules of about twenty-two nucleotides and have crucial roles in both healthy and pathological cells. In fact, they engage in various intra- and intercellular mechanisms like proliferation, apoptosis, and autophagy. Therefore, an incorrect function of miRNAs participates in the onset of different diseases like neoplasms [1,2]. The scientific literature shows the correlation between sex differences and miRNA expression. The latter is a complex relationship that deeply influences the modality of presentation of distinct types of tumors and organism response to treatment (Table 2). In addition, it is not possible to state that a single miRNA has the power to determine epigenetic changes in the human genome. There is evidence of the fact that clusters of miRNAs are coexpressed in a specific type of cancer and act in a proproliferative or antiproliferative sense, radically modifying the tumoral microenvironment and activating particular cells of the immune system against cancer [86,87,88,89,90,91,92,93,94,95]. This is the case of papillary thyroid cancer [30], in which miR-221, miR-222, miR-181a and miR-181b, and miR-21 are present at high levels of hepatocellular carcinoma [42,43] in which the family of miR-371-373, expressed in stem cells during embryogenesis and then normally reduced after cell differentiation, is upregulated in adult patients, especially in men and promotes cell proliferation and neoangiogenesis [96,97,98,99,100,101,102,103,104,105,106,107,108,109,110,111]. MiRNAs expression can be modified by the action of drugs. For example, in gastric cancer, bicalutamide single or combined with 50 nM of dihydrotestosterone causes an important decrease in the expression of miR-125 and, consequently, apoptosis of neoplastic cells occurs [48].

Further fields of study could be the differential analysis between circulating miRNAs and organ-specific miRNAs. Circulating miRNAs, also called cell-free miRNAs, are in the body fluids and serve as laboratory biomarkers for both pathological and physiological conditions like cancer, organ damage, and pregnancy. After apoptosis or necrosis, intracellular miRNAs are rejected in the extracellular environment and become circulating miRNAs, which can be encapsulated in microvesicles like exosomes and shedding vesicles enveloped by a phospholipid bilayer or tied to high-density lipoproteins (HDL) molecules. Ninety-ninety percent of them are constituted by argonaute-binding (AGO-binding) miRNAs, whose function is still uncertain. It is well known that AGO proteins are fundamental catalytic molecules of the “RNA-induced silencing complex” (RISC) and have crucial roles in the regulation of gene expression. In fact, during this mechanism, miRNA ties to an AGO protein that gives it stability. Evidence showed that circulating miRNA profiles and concentrations differ from one body fluid to another. In fact, in urine, there is the lowest concentration compared with the highest in serum. This condition could be a consequence of cell disruption due to the coagulation phenomenon [113,114,115,116]. As far as organ-specific miRNAs are concerned, in the liver the most common miRNA is miR-122, which regulates the response to stress, metabolism, and maintenance of hepatic phenotype; if the hepatic microenvironment changes, for example in alcohol-induced diseases or infections, levels of miR-122 are deregulated. Thus, it can be considered a liver-specific miRNA. In the case of heart damage, miR-208 is released into serum so it could be used as a novel biomarker for heart injury [117,118,119]. Indeed, it is possible that a different profile is present in the two sets of miRNAs and the circulating miRNAs do not necessarily represent a reliable reproduction of the situation in the organ affected by the neoplasia.

Obviously, further studies are necessary in order to clarify mechanisms of action and targets of the various miRNAs with the aim of identifying some of these small noncoding molecules as possible cancer-specific biomarkers and to correlate their levels of expression with tumoral aggressivity, prognosis, and response to therapy parameters.

## Figures and Tables

**Figure 1 ijms-24-11544-f001:**
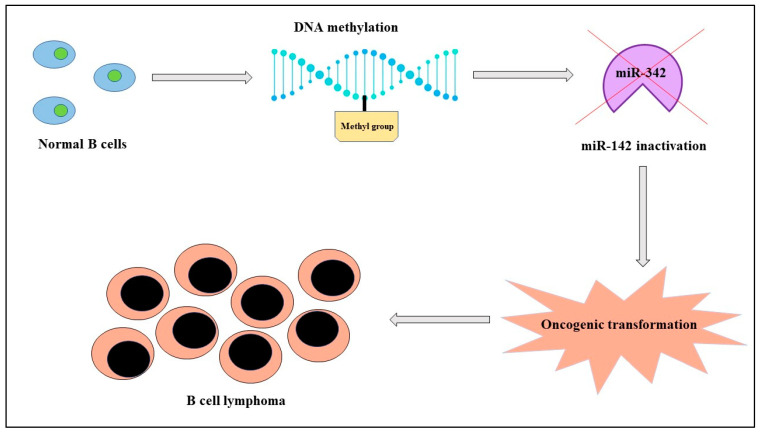
In B-cell lymphomas, the activity of miR-142, localized in the EVL gene of chromosome 14, is blocked by the hypermethylation of DNA, leading to cancerogenesis.

**Figure 2 ijms-24-11544-f002:**
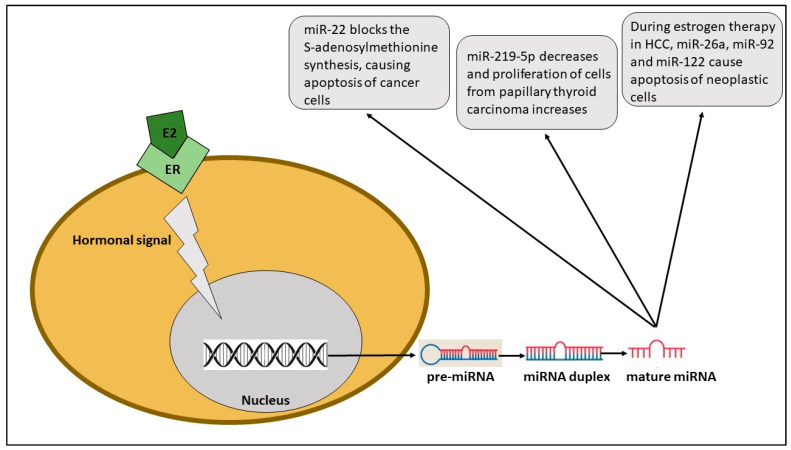
Estrogen (E2) ties to the estrogen receptor (ER), causing a signal cascade to the nucleus, leading to mature miRNA, which has different effects on cancer cells.

**Table 1 ijms-24-11544-t001:** MiRNAs involved in apoptosis and autophagy regulation of several types of cancer.

Cancer	miRNAs	UP/DOWN Regulated	Mechanism of Action	Onset/Prognosis/Response to Therapy Involvement	References
B-cell lymphoma	miR-142	DOWN	Altered migration of cells	Poor prognosis	[22]
Papillary thyroid cancer	miR-21 miR-26a miR-181a miR-181b miR-219 miR-221 miR-222 miR-245	UP	Modulation in protein p27Kip1 expression	Onset	[30]
Hepatocellular carcinoma	miR-371a-5p	UP	Transition from G to S phase of cell cycle	Onset	[41]
Colorectal cancer	miR-16 miR-22 miR-142-3p	miR-22 UP; miR-16 and miR-142-3p DOWN	Inhibition of autophagy (miR-16, miR-142-3p) and inhibition of estrogen activity	Onset (miR-16, miR-142-3p) and better response to therapy (miR-22)	[8]
Gastric cancer	miR-125	UP	Block of apoptosis	Onset	[48]
Lung cancer	miR-143 miR-145 miR-153-3p	UP	Apoptosis induction	Good prognosis	[22]
Melanoma	miR-23a miR-221 miR-222	UP	Block of cell proliferation	Onset	[8]
Breast cancer	miR-17 miR-21 miR-124	UP	BRCA1 inactivation	Onset	[80,81]
Glioblastoma	hsa-miR-1909-5p hsa-let-7c-5p miR-206-5p	hsa-miR-1909-5p and hsa-let-7c-5p UP; miR-206-5p DOWN	Promotion of cell migration and invasion (hsa-miR-1909-5p, hsa-let-7c-5p) and modulation of apoptosis (miR-206-5p)	Onset	[85]

**Table 2 ijms-24-11544-t002:** Completed clinical trials about miRNAs in different types of cancer (clinicaltrials.gov accessd on July 2023) [112].

Study Title	Conditions	Study Type	Inclusion Criteria	Sex	Age	NCT Number
Studying Genes in Samples From Younger Patients With Ovarian or Testicular Sex Cord Stromal Tumors	Childhood Germ Cell Tumor; Leydig Cell Tumor; Ovarian Cancer	Observational	Fixed and frozen tissue samples from the ATBR01 B1 tissue bank and from the International Pleuropulmonary Blastoma Registry, Children’s Hospital of Boston, and Massachusetts General Hospital	All	up to 120 Years	NCT01572467
Exosomal microRNA in Predicting the Aggressiveness of Prostate Cancer in Chinese Patients	Prostate Cancer	Observational	Patients clinically diagnosed to have localized Prostate Cancer and planned for radical prostatectomy; no prior systemic therapy for Prostate Cancer used, including hormonal or chemotherapy	Male	45 Years and older	NCT03911999
The Long Noncoding MALAT1 as a Potential Salivary Diagnostic Biomarker in Oral Squamous Cell Carcinoma Through Targeting mi RNA 124	Oral Cancer Biomarkers	Observational	Patients with Oral Squamous Cell Carcinoma and healthy controls of both sexes	All	Child, Adult, Older Adult	NCT05708209
Elucidating the Genetic Basis of the Pleuropulmonary Blastoma (PPB) Familial Cancer Syndrome	Pleuropulmonary Blastoma; Cystic Nephroma; Sertoli-Leydig Cell Tumor of Ovary	Observational	Patients diagnosed with pleuropulmonary blastoma, cystic nephroma, embryonal rhabdomyosarcoma of uterine cervix, ovarian Sertoli–Leydig tumor or gynandroblastoma, pineoblastoma, pituitary blastoma, nasal chondromesenchymal hamartoma, medulloepithelioma, Wilms tumor, germline or mosaic DICER1 mutation	All	1 Day to 95 Years	NCT00565903
STI.VI. Study: How to Improve Lifestyles in Screening Contexts	Lifestyle Risk Reduction; Weight Changes, Body; Breast Cancer	Interventional	49- to 55-year-old women invited to mammography screening; 58- to 61-year-old people (both sexes) invited to colorectal screening	All	49 Years to 61 Years	NCT03118882

## Data Availability

Not applicable.

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
