# Peer review of "Gender Differences and miRNAs Expression in Cancer: Implications on Prognosis and Susceptibility"

_ijms, 2023, doi:10.3390/ijms241411544_

Round 1

Reviewer 1 Report

Summary: 

 In this review paper, authors reviewed the correlation among miRNAs expression, sex, and cancer in both solid tumors and hematological malignancies. This review covered common cancer types include Hematological malignancies, Thyroid cancer, Hepatocellular carcinoma, Colorectal cancer, Gastric cancer, Lung cancer, Melanoma, Breast cancer and Glioblastoma. This paper commented on the correlation between miRNA expression and disease onsite, prognosis, or response to therapy. Overall, this is a very important topic, with potential broad interest from targeted audience of IJMS, and has the potential to provide some insights into cancer diagnosis and therapy. One thing to note is that, references seems missing in quite a few cases and need to be added back in. I noted a few places where reference seems missing, but probably didn’t cover all. More detailed comments please see below.

Detailed comments:

1. Page 2 line 65, reference needed

2. Page 2 line 76, please define “oncoMirs”

3. Page 5 line 170-174, confusing long sentence, please consider break them into multiple sentences.

4. Page 6 line 204, reference seems missing?

5. Page 7 line 259, reference missing

6. Page 10 line 357, reference seems missing

7. Page 11 Table 1, It would be super helpful and very informative if the author could add in a column describing what role miRNA play in associated cancer types. Also, it would be great if authors could note if the miRNA is related to disease onsite, prognosis, or response to therapy.

The English language used is a little too colloquial in some cases, for example the frequent use of "thanks to" and "anyways".  Also in some cases, the authors like to use very long sentences with complexed structures which are hard to understand.

Author Response

Dear reviewer, I revise the paper accordingly to your suggestions.

Giuseppe Murdaca

Reviewer 2 Report

1. It is not described by what criteria studies were selected for review. In general, there is no Methods section.

2. The emphasis in the title is on gender differences, it seems to me more correct to call them sexual, but the information in the article is extremely chaotic. So, I would like to see a summary table that would summarize the data on miRNAs in different cancers, depending on sex, with a description of the number of study participants, inclusion criteria, etc. It is stated that the prognosis of the disease is assessed using miRNA, but specific data are also not given: specific criteria for good or poor prognosis are needed in relation to gender.

3. There is very little information in Table 1: are these miRNAs downregulated or upregulated and for which sex is this observed?

Author Response

Dear reviewer, I revised the paper accordingly to your suggestions.

Giuseppe Murdaca

Round 2

Reviewer 2 Report

I have no more comments on the article. I believe that in its present form the manuscript can be recommended for publication.

Author Response

Dear reviewer, many thanks for you endorsement.

Giuseppe Murdaca
